# Clinical features and severe acute respiratory syndrome-coronavirus-2 structural protein-based serology of Mexican children and adolescents with coronavirus disease 2019

Karen Cortés-Sarabia[1], Armando Cruz-Rangel[2], Alejandro Flores-Alanis[3], Marcela Salazar-García[4,5], Samuel Jiménez-García[6], Griselda Rodríguez-Martínez[5], Juan Pablo Reyes-Grajeda[2], Rosa Isela Rodríguez-Téllez[7], Genaro Patiño-López[7], Israel Parra-Ortega[8], Oscar Del Moral-Hernández[9], Berenice Illades-Aguiar[6], Miguel Klünder-Klünder[10], Horacio Márquez-González[11], Adrián Chávez-López[12], Victor M. Luna-Pineda[5,7]*

1 Laboratorio de Inmunobiología y Diagnóstico Molecular, Facultad de Ciencias Químico Biológicas, Universidad Autónoma de Guerrero, Guerrero, México, 2 Laboratorio de Bioquímica de Enfermedades Crónicas, Instituto Nacional de Medicina Genómica, Mexico City (Ciudad de México), México, 3 Departamento de Microbiología y Parasitología, Facultad de Medicina, Universidad Nacional Autónoma de México, Mexico City (Ciudad de México), México, 4 Laboratorio de Biología del Desarrollo y Teratogénesis Experimental, Hospital Infantil de México Federico Gómez, Mexico City (Ciudad de México), México, 5 Laboratorio de Investigación en COVID-19, Hospital Infantil de México Federico Gómez, Mexico City (Ciudad de México), México, 6 Laboratorio de Biomedicina Molecular, Facultad de Ciencias Químico Biológicas, Universidad Autónoma de Guerrero, Guerrero, México, 7 Unidad de Investigación en Inmunología y Proteómica, Hospital Infantil de México Federico Gómez, Mexico City (Ciudad de México), México, 8 Laboratorio Central, Hospital Infantil de México Federico Gómez, Mexico City (Ciudad de México), México, 9 Laboratorio de Virología, Facultad de Ciencias Químico Biológicas, Universidad Autónoma de Guerrero, Guerrero, México, 10 Subdirección de Gestión de la Investigación, Hospital Infantil de México Federico Gómez, Mexico City (Ciudad de México), México, 11 Investigación Clínica, Hospital Infantil de México Federico Gómez, Mexico City (Ciudad de México), México, 12 Departamento de la Unidad de Terapia Intensiva Pediátrica, Hospital Infantil de México Federico Gómez, Mexico City (Ciudad de México), México

* luna.pineda@hotmail.com

**Data Availability Statement:** The individual patient data used to generate the findings of this study are

## Abstract

Severe acute respiratory syndrome (SARS)-coronavirus (CoV)-2 infection in children and adolescents primarily causes mild or asymptomatic coronavirus disease 2019 (COVID-19), and severe illness is mainly associated with comorbidities. However, the worldwide prevalence of COVID-19 in this population is only 1%–2%. In Mexico, the prevalence of COVID-19 in children has increased to 10%. As serology-based studies are scarce, we analyzed the clinical features and serological response (SARS-CoV-2 structural proteins) of children and adolescents who visited the Hospital Infantil de México Federico Gómez (October 2020–March 2021). The majority were 9-year-old children without comorbidities who were treated as outpatients and had mild-to-moderate illness. Children aged 6–10 years and adolescents aged 11–15 years had the maximum number of symptoms, including those with obesity. Nevertheless, children with comorbidities such as immunosuppression, leukemia, and obesity exhibited the lowest antibody response, whereas those aged 1–5 years with heart disease had the highest levels of antibodies. The SARS-CoV-2 spike receptor-binding

available on Harvard Dataverse (https://dataverse.harvard.edu) upon acceptance of the manuscript. In addition, the format of the consent and settlement letters was included in this data repository. DOI access is https://doi.org/10.7910/DVN/DQNW2J.

**Funding:** This research article was funded by Public Federal Funds from Hospital Infantil De Mexico Federico Gomez with grant numbers HIM-2020-029 (VL), HIM-2020-060 (VL), and HIM-2021-007 (VL).

**Competing interests:** The authors have declared that no competing interests exist.

domain-localized peptides and M and E proteins had the best antibody response. In conclusion, Mexican children and adolescents with COVID-19 represent a heterogeneous population, and comorbidities play an important role in the antibody response against SARS-CoV-2 infection.

## Introduction

In December 2019, a new coronavirus (CoV) was identified in patients with atypical pneumonia in Wuhan, Hubei province, China [1]. This virus was defined as severe acute respiratory syndrome (SARS)-CoV-2, and the disease caused by this virus was named coronavirus disease 2019 (COVID-19) by the World Health Organization. To date, SARS-CoV-2 has infected more than 217 million people and caused more than four million deaths worldwide (https://coronavirus.jhu.edu/map.html). Based on the symptoms, COVID-19 can be classified as asymptomatic, mild, or severe illness, and the most common symptoms include fever, cough, fatigue, exhaustion, chest pain, headache, diarrhea, and vomiting [2,3]. Furthermore, clinical complications such as acute respiratory distress syndrome, arrhythmia, shock, acute kidney injury, acute heart injury, and liver dysfunction have been noted in patients with severe COVID-19 [4]. COVID-19 cases have presented differences in age groups: the highest incidence was reported in elderly people at the beginning of the pandemic, whereas subsequently, there was an increase in the number of cases in people aged 20–49 years [5]. In contrast, young people, including children and adolescents, showed the lowest prevalence of infection, which could be due to less frequent or asymptomatic infection; however, they may spread the virus to other children or adults with comorbidities, although the exact mechanism is not completely understood [3,6,7].

In Mexico, according to data from the Secretary of Health, more than three million cases of COVID-19 and more than 300,000 deaths have been confirmed since the first positive patient was reported. Among the COVID-19 cases, more than 300,000 corresponded to children and adolescents (~10%), with more than 1000 deaths since February 27, representing an increase in both incidence and mortality in the age groups of 10–14 (>74,000) and 15–19 (>152,000) years (https://coronavirus.gob.mx/datos/). The incidence of COVID-19 in children in Mexico represents a 5- to 10-fold increase compared with that in countries such as China, the USA, Italy, and Spain (1%–2%) [8–11]. Although the factors contributing to the increased incidence of COVID-19 have not been systematically analyzed, it has been suggested to be associated with the success of SARS-CoV-2 vaccination in the adult population.

The clinical manifestations of SARS-CoV-2 infection in children are distinct from those in adults, who rarely exhibit severe respiratory symptoms and are often asymptomatic [12]. Therefore, serological testing of asymptomatic or infected children is important to evaluate the immune response; antibodies not only play a protective role in limiting infection and preventing future reinfection, but they are also important for the rapid identification of infection and treatment [12,13]. The reported seroprevalence rate of SARS-CoV-2 infection in noninfected children is 1.2% for IgM and 0.6% for IgG, whereas the kinetics of antibody response demonstrated an initial seroprevalence of 6.9% and 7.66% after 62 days, respectively [14,15]. The majority of serological tests developed for the detection of IgM and IgG antibodies are targeted to the spike (S) and nucleocapsid (N) structural proteins of the SARS-CoV-2 virus in serum or plasma samples; however, other structural proteins should be included, such as membrane (M) and envelope (E) proteins [16]. Contrary to reverse transcription-polymerase chain

reaction (RT-qPCR), antibody testing has the potential to detect previous asymptomatic/mildly symptomatic infection and is not dependent on coinciding with active infection [17,18].

Despite the global spread of SARS-CoV-2, the epidemiological and clinical patterns of COVID-19 in children and adolescents are largely unclear. Hence, the proportion of asymptomatic children and the primary symptoms associated with children and adolescents with COVID-19 are not well established [19,20]. In addition, reports of antibody response in children and adolescents are limited and are only focused on the analysis of SARS-CoV-2 S and N proteins as antigens [21,22]. Until October 2021, in Hospital Infantil de México Federico Gómez (HIMFG), more than 7800 children and adolescents have been evaluated for COVID-19 using diagnostic RT-qPCR, among which only 755 tested positive (unpublished). Therefore, a comprehensive study evaluating both the clinical features and antibody response to structural proteins in Mexican children and adolescents with previous SARS-CoV-2 infection is necessary to obtain a better understanding of the factors associated with the infection, hospitalization, severity of illness, and comorbidities. Hence, we analyzed the clinical features and serological response to SARS-CoV-2 structural proteins in 100 children and adolescents with COVID-19 who visited HIMFG from October 2020 to March 2021.

## Materials and methods

### Patient population and definition

Children and adolescents with COVID-19 were defined as those who tested positive for SARS-CoV-2 using RT-qPCR assays for nasopharyngeal samples performed at the Central Laboratory of HIMFG between October 2020 and March 2021. Children were defined as those aged <11 years, and adolescents were defined as those aged 11–18 years.

SARS-CoV-2-positive patients were categorized as follows: 1) asymptomatic (absence of symptoms), 2) mild (signs and symptoms with outpatient management), 3) moderate (mild illness but with hospital management), and 4) severe COVID-19 (patients who were admitted to the intensive care unit). The sex, age, symptoms, severity of illness, admission service, comorbidities, and clinical course of each patient were obtained from medical records.

The symptoms were defined as follows: 1) fever: body temperature higher than normal (98.6˚F [37˚C]); 2) sudden onset of symptoms: referring to those who developed symptoms over a short period of time; 3) cough: a reflex action to clear the airways of mucus and irritants; 4) odynophagia: painful swallowing; 5) dyspnea: shortness of breath or breathlessness; 6) irritability: mood characterized by a susceptibility to experience negative affective states; 7) diarrhea: loose, watery, and possibly more frequent bowel movements; 8) chest pain: pain associated with deep breathing, coughing, or sneezing, and inflammation of the lungs; 9) shivers: rapid contraction and release of the muscles; 10) headache: pain experienced in the face, head, or neck; 11) myalgia: muscle pain; 12) arthralgia: joint pain; 13) attack on the general state: associated with three clinical signs, i.e., anorexia, fatigue, and weight loss; 14) rhinorrhea: free discharge of a thin nasal mucus fluid; 15) polypnea: rapid breathing; 16) vomiting: act of ejecting the contents of the stomach through the mouth as a result of involuntary muscular spasms of the stomach and esophagus; 17) stomachache: pain in the stomach or abdomen; 18) conjunctivitis: inflammation of the conjunctiva; 19) cyanosis: bluish-purple discoloration of the skin and mucous membranes generally resulting from oxygen deficiency in the blood; 20) anosmia: loss of the sense of smell; and 21) dysgeusia: distortion of the sense of taste.

### Ethical statement

This study was conducted at HIMFG and was reviewed and approved by the Research Committee, Ethics Committee, and Biosecurity Committee (HIM-2020-029 and HIM-2021-007

grants). Blood samples were collected according to the ethics and biosafety protocols published in the standardized guidelines for the laboratory and epidemiological surveillance of viral respiratory diseases from the Departamento de Vigilancia Epidemiologica (https://coronavir us.gob.mx/wp-content/uploads/2020/04/Lineamientodevigilancia_epidemiologica_de_enfer medad_respiratoria-_viral.pdf). The parents accepted in written format the use of blood samples by signing the informed consent and settlement letters and the names of the patients were anonymized. The consent and settlement letters are available on https://doi.org/10.7910/DVN /DQNW2J. In total, 10 serum samples were collected from each COVID-19-positive and -negative patient with informed consent. Positive samples were collected from infected health workers (confirmed using RT-qPCR), and negative samples were collected from patients who assisted in the routine analysis at HIMFG. The presence of SARS-CoV-2-specific antibodies in sera samples of COVID-19-positive and -negative patient were qualitatively evaluated using Novel Coronavirus (2019-nCoV) IgG/IgM Test Kit (Genrui Biotech). The plasma and serum samples were aliquoted and stored at −20˚C until serological testing.

## Purification of SARS-CoV-2 structural proteins

SARS-CoV-2 S HexaPro plasmid (Addgen) was transfected into the human embryonic kidney cell line HEK293F, and the cells were cultured in 293 FreeStyle expression medium (Life Technologies) at 37˚C with 5% $CO_2$. After 48 h, the supernatant was recovered, centrifuged, filtered, and purified using immobilized metal affinity chromatography (IMAC) in native conditions. The resulting protein had six prolines substituted, a substitution of "GSAS" (furin cleavage site), and the addition of a C-terminal fold on the trimerization motif with an eight-histidine tag that generates a prefusion-stabilized SARS-CoV-2 S protein [23]. The SARS-CoV-2 S1 subunit was obtained for the elimination of residues 681 to 1208 from SARS-CoV-2 S HexaPro plasmid (5′EcoRI and 3′BamHI digest), and the resulting plasmid was named pS1-LC. The S1 subunit included residues 28 to 680 with a C-terminal fold on the trimerization motif and eight histidine tag. The SARS-CoV-2 receptor-binding domain (RBD) was amplified from the cDNA of a SARS-CoV-2 clinical isolate and cloned in pLATE-31 according to the protocol of the aLICator Ligation Independent Cloning and Expression System (Thermo Scientific). pLATE-RBD with a C-terminal six histidine tag was transformed into Escherichia coli strain Rosetta and purified using IMAC in native conditions. The SARS-CoV-2 N protein was also amplified from the cDNA of a SARS-CoV-2 clinical isolate but was cloned in pLATE-51 containing an N-terminal six histidine tag by E. coli strain Rosetta induction and IMAC purification in native conditions. The SARS-CoV-2 E and M proteins were also cloned in pLATE-51 and transformed into E. coli strain Rosetta, but these were purified using IMAC in denaturalized conditions. The SARS-CoV-2 structural proteins were quantified using the Quick Start™ Bradford Protein Assay Kit (BioRad) and visualized using Coomassie staining.

## SARS-CoV-2 S peptides

The multiantigenic peptide 8 (MAP8) format allows the synthesis of peptides with a length of ≤15 residues. Peptide synthesis was performed by PepMic (http://www.pepmic.com/), and each peptide was dissolved to a final concentration of 1 mg/mL. Five peptides located in the RBD of the S protein selected using in silico analysis from the primary amino acid sequence and synthetized in MAP8 were used as the antigen. The amino acid sequences of these peptides located in the S1 domain are SNNLDSKVGGNY, RLFRKSNLKPFE, ISTEIYQAGST, YGFQPTNGVGYQ, and GPKKSTNLVKNK. The purity of peptides was verified using HPLC-Reversed Phase (Luna 3u C18[2] column, Phenomenex Inc.), and the identity of peptides was determined using mass spectrometry (manuscript under review).

## Standardization of indirect enzyme-linked immunosorbent assay (ELISA)

After purification and quantification of the recombinant SARS-CoV-2structural proteins, we performed the standardization procedure for their use as antigens to detect antibodies in serum samples collected from children and adolescents. For standardization, we used serum samples collected from SARS-CoV-2-positive patients (confirmed using RT-qPCR) as positive controls, with at least 15 days after the end of symptoms, and samples collected from SARS-CoV-2-negative patients (negative RT-qPCR result) were used as negative controls. We performed double serial dilution of each antigen (from 1 to 0.01 µg/mL), which was subsequently adjusted, with the following parameters: serum dilution (1:25 to 1:500), dilution buffer (PBS, 1 and 3% skimmed milk) and incubation time (15, 30, and 45 min). Before use, each secondary antibody (anti-IgG whole molecule [wm] and anti-IgG γ-specific) was titrated. The cutoff value was calculated considering the mean ± 3 standard deviation of the absorbances of negative controls, and samples with absorbances of >0.150 for anti-human wm IgG and >0.180 for anti-human γ-specific IgG were considered positive.

## Serological evaluation of SARS-CoV-2 structural proteins and peptides

Indirect ELISA was performed to detect antibodies using both the recombinant proteins derived from SARS-CoV-2 structural proteins (S, RBD, N, M, and E proteins) and a mixture of five peptides in the RBD-located MAP8 format from the S protein as antigens. Microtiter plates (Sigma-Aldrich) were coated with 100 µL/well of individual recombinant structural proteins or peptides (equal to 20 ng/peptide) at a final concentration of 0.1 µg/mL in a coating buffer (50 mM $Na_2CO_3$/$NaCO_3H$, pH 9.6). The plates were incubated for 1 h at 37˚C and then blocked for 30 min at 37˚C with 200 µL of 5% skimmed milk diluted in phosphate-buffered saline (PBS)-Tween 20 (0.05%). The serum or plasma samples (in duplicate) were diluted 1:250 (structural proteins) or 1:50 (peptides) for detection of total antibodies, whereas a dilution of 1:50 (both antigens) was used for the specific detection of IgG in PBS (pH 7.2) at 100 µL/well with incubation for 30 min and 1 h, respectively. The plates were then incubated with 100 µL of either anti-human wm IgG (Sigma-Aldrich; 1:8000 dilution) or anti-human IgG (γ-chain specific; 1:2000 dilution) coupled to horseradish peroxidase at 37˚C for 30 min and 1 h, respectively. After every step, the plates were washed three times with 200 µL/well of 0.05% PBS-Tween 20 for 5 min. The enzymatic reaction was developed using *o*-phenylenediamine dihydrochloride (Sigma-Aldrich) and stopped using 2 N $H_2SO_4$. The resulting optical density was measured at 492 nm using a microplate reader.

## Statistical analysis

The results of the comparison of antibody titers (wm and γ-chain IgGs) between demographic (age and sex) and clinical variables (symptomatology and patient status) were reported as medians (maximum and minimum). Values were evaluated for normality and outliers using the Shapiro–Wilk test. The nonparametric Kruskal–Wallis test was conducted, and Dunnett's test was performed as a post hoc analysis test; $p < 0.05$ was considered statistically significant. All statistical analyses were performed using RStudio version 3.2.2, and the graphic representation was conducted using GraphPad Prism version 9.0.0 for Windows (GraphPad Software, San Diego, California, USA; www.graphpad.com).

## Results

One-hundred children and adolescents with COVID-19 were enrolled in this study. All samples were collected at HIMFG from October 2020 to March 2021. For the follow-up analysis,

clinical data were obtained from medical records, and blood samples (serum or plasma) were acquired 20–30 days after the onset of symptoms for serological analysis.

## Clinical features of the children and adolescents with COVID-19

The 100 patients enrolled in this study accounted for 13.2% (100/755 cases) of all diagnosed patients at HIMFG; 56 were boys and 44 were girls, with a median age of 9 years (range: 0–18 years). Of these patients, 61 were outpatients and 39 were hospitalized. The hospitalized patients were admitted to the following departments: Pediatric Emergency (35%), Neonatal/ Pediatric Intensive Care Unit (2%), Infectiology (1%), and Internal Medicine (1%), and the ambulatory cases were attended by external consultation. In adults, comorbidities are associated with clinical complications and low immune response. To perform a more in-depth analysis, we included these clinical features in the study population. We found that 37% of the children and adolescents with COVID-19 had comorbidities, such as cardiovascular disease (9%), leukemia (7%), obesity (6%), immunosuppression/HIV-AIDS (6%), chronic kidney failure (2%), asthma (2%), and diabetes (2%), whereas only 3% had unspecified comorbidities (Table 1). The most frequent diseases attended to at HIMFG were acute lymphoblastic leukemia and cardiovascular disease; however, there was no significant difference between demographic status and comorbidities in this population.

The major clinical features associated with SARS-CoV-2 infection are symptoms, although most children and adolescents remain asymptomatic. In the analyzed population, the most common clinical feature was fever (45%), followed by a sudden onset of symptoms (29%), cough (28%), headache (24%), attack on the general state (22%), odynophagia (20%), vomiting

**Table 1. Demographic and clinical features of 100 Mexican children and adolescents with COVID-19.**

| Demographic and clinical features | (%) |
|---|---|
| Age | Median 9 years; range 0–18 years |
| Sex | Female 44/Male 56 |
| Patient status | Hospitalized 39 |
| Pediatric Emergency | 35 |
| Neonatal/pediatric Intensive Care Unit | 2 |
| Infectiology | 1 |
| Internal Medicine | 1 |
| External consultation | Ambulatory 61 |
| Comorbidity | |
| Without comorbidity | 63 |
| Cardiovascular disease | 9 |
| Leukemia | 7 |
| Obesity | 6 |
| Immunosuppression/HIV-AIDS | 6 |
| Chronic kidney failure | 2 |
| Asthma | 2 |
| Diabetes | 2 |
| Unspecified comorbidities | 3 |
| Severity of illness | |
| Asymptomatic | 19 |
| Mild | 42 |
| Moderate | 36 |
| Severe | 3 |

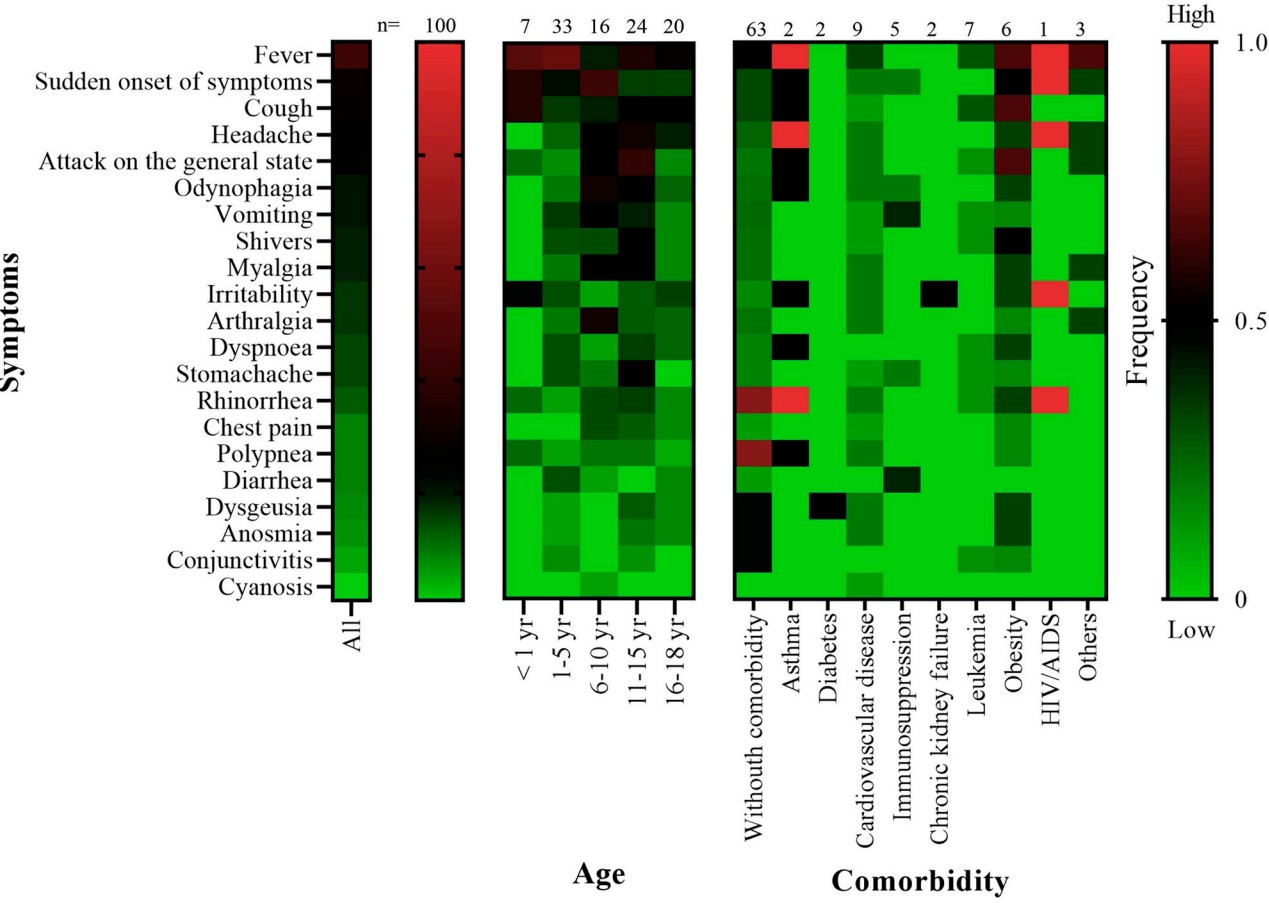

**Fig 1. Distribution of symptoms, age, and comorbidities in Mexican children and adolescents with COVID-19.** Heat map showing the frequency of symptoms in Mexican children and adolescents with COVID-19 (left panel) and their age and the comorbidities. The total number (n) of patients with COVID-19 is shown on the top of the figure, and the number of patients corresponding to each age category or presenting any comorbidity is displayed. The frequency scale of the variables is shown at the right. The color map was prepared with a double gradient, where the largest value is represented in red (100%), baseline value is represented in black (50%), and smallest value is represented in green (0%).

(19%), shivers (19%), myalgia (19%), irritability (17%), arthralgia (17%), dyspnea (15%), stomachache (15%), and rhinorrhea (13%), whereas chest pain (9%), polypnea (9%), diarrhea (9%), dysgeusia (8%), anosmia (7%), conjunctivitis (5%), and cyanosis (1%) were observed less frequently (Fig 1).

## Patient status and severity of illness in Mexican children and adolescents with COVID-19

The HIMFG is a tertiary care hospital for children and adolescents. During the current pandemic, it has been listed as a COVID-19 hospital in Mexico. From October 2020 to March 2021, this COVID-19 hospital attended to ambulatory cases of 61 children and adolescents, corresponding to 33.3% of children aged 1–5 years and 21.6% of adolescents aged 11–15 years. The hospitalized patients (39%) included the aforementioned age groups (Table 2). Most of them were admitted to Pediatric Emergency (89.5%), followed by the Neonatal Intensive Care Unit (2.6%), Internal Medicine (2.6%), Infectology (2.6%), and the Intensive Care Unit (2.6%).

**Table 2. Distribution of patient status and severity of illness in Mexican children and adolescents with COVID-19.**

| Age group (years) | Total | Patient status (%) | | Severity of illness (%) | | | |
|---|---|---|---|---|---|---|---|
| | | Ambulatory | Hospitalized | Asymptomatic | Mild | Moderate | Severe |
| | n = 100 | n = 61 | n = 39 | n = 19 | n = 42 | n = 36 | n = 3 |
| <1 | 7 | 5 (8.3) | 2 (5.1) | 1 (5.3) | 4 (9.5) | 2 (5.6) | 0 (0) |
| 1–5 | 33 | 20 (33.3) | 13 (33.3) | 6 (31.6) | 14 (33.3) | 12 (33.3) | 1 (33.3) |
| 6–10 | 16 | 10 (16.6) | 6 (15.4) | 2 (10.5) | 8 (19.0) | 6 (16.7) | 0 (0) |
| 11–15 | 24 | 13 (21.6) | 11 (28.2) | 5 (26.3) | 8 (19.0) | 9 (25.0) | 2 (66.7) |
| 16–18 | 20 | 12 (20) | 7 (17.9) | 5 (26.3) | 8 (19.0) | 7 (19.4) | 0 (0) |

Further classification was made based on patient status and symptoms. Thus, patients were categorized as having asymptomatic (19%), mild (42%), moderate (36%), and severe (3%) COVID-19. Asymptomatic patients primarily included children aged 1–5 years (31.6%) and adolescents aged 11–18 years (52.6%), whereas those with mild COVID-19 were mainly distributed among individuals aged 1–5 years. Moderate illness, similar to the asymptomatic condition, was predominant in children aged 1–5 years (33.3%) and adolescents aged 11–15 years (25%). Severe COVID-19 was only observed in the age groups of 1–5 years (33.3%) and 11–15 years (66.7%). Interestingly, the lowest frequency of symptoms was observed in newborns and infants (<1 year), whereas most of the children and adolescents were discharged with satisfactory prognosis, and only two adolescents (11 and 15 years) were intubated. Severe illness was associated with obesity as a comorbidity (Table 2).

## SARS-CoV-2 structural protein-based serology of Mexican children and adolescents with COVID-19

As the major SARS-CoV-2 structural proteins are S (trimer and RBD), N, M, and E, we cloned, expressed, and purified these proteins for use as antigens in the serological analysis. Five RBD-localized peptides (RBD peptides) in the MAP8 format were included in this analysis. The wm IgG determination and use of RBD peptides as the antigen showed the highest absorbance (Abs, $OD_{492}$), with a median of 0.833 (maximum 1.726; minimum 0.166), compared with S, RBD, N, E, and M proteins ($p \leq 0.0113$; Fig 2a).

To determine whether there was a relationship between the age of serum donors and IgG antibodies, we performed a comparison between wm IgG antibodies and age. The highest median of Abs corresponded to the age group of 1–5 years when using S (median 0.756; maximum 1.416; minimum 0.230), RBD (median 0.723; maximum 1.190; minimum 0.136), RBD peptides (median 1.150; maximum 1.703; minimum 0.290), N (median 0.636; maximum 1.443; minimum 0.110), M (median 0.643; maximum 1.463; minimum 0.116), or E (median 0.623; maximum 1.556; minimum 0.110) proteins as the antigen (Fig 2b).

We then analyzed whether sex, severity of illness, patient status, or presence of comorbidities had an effect on the antibody response. None of these factors produced any significant differences in the number of IgG antibodies when any of the aforementioned antigens were used. RBD peptides showed the highest median of Abs when a comparison was made among the SARS-CoV-2 structural proteins (Fig 2c and 2d). The observed median Abs values for RBD peptide were 0.761 (maximum 1.701; minimum 0.275) for female and 0.868 (maximum 1.728; minimum 0.168) for male (Fig 2c) patients. When the severity of illness was analyzed, the observed median Abs values were 0.766 (maximum 1.693; minimum 0.286) for asymptomatic, 0.820 (maximum 1.726; minimum 0.160) for mild illness, 0.786 (maximum 1.506; minimum 0.253) for moderate, and 1.040 (maximum 1.693; minimum 0.273) for severe illness (Fig 2d).

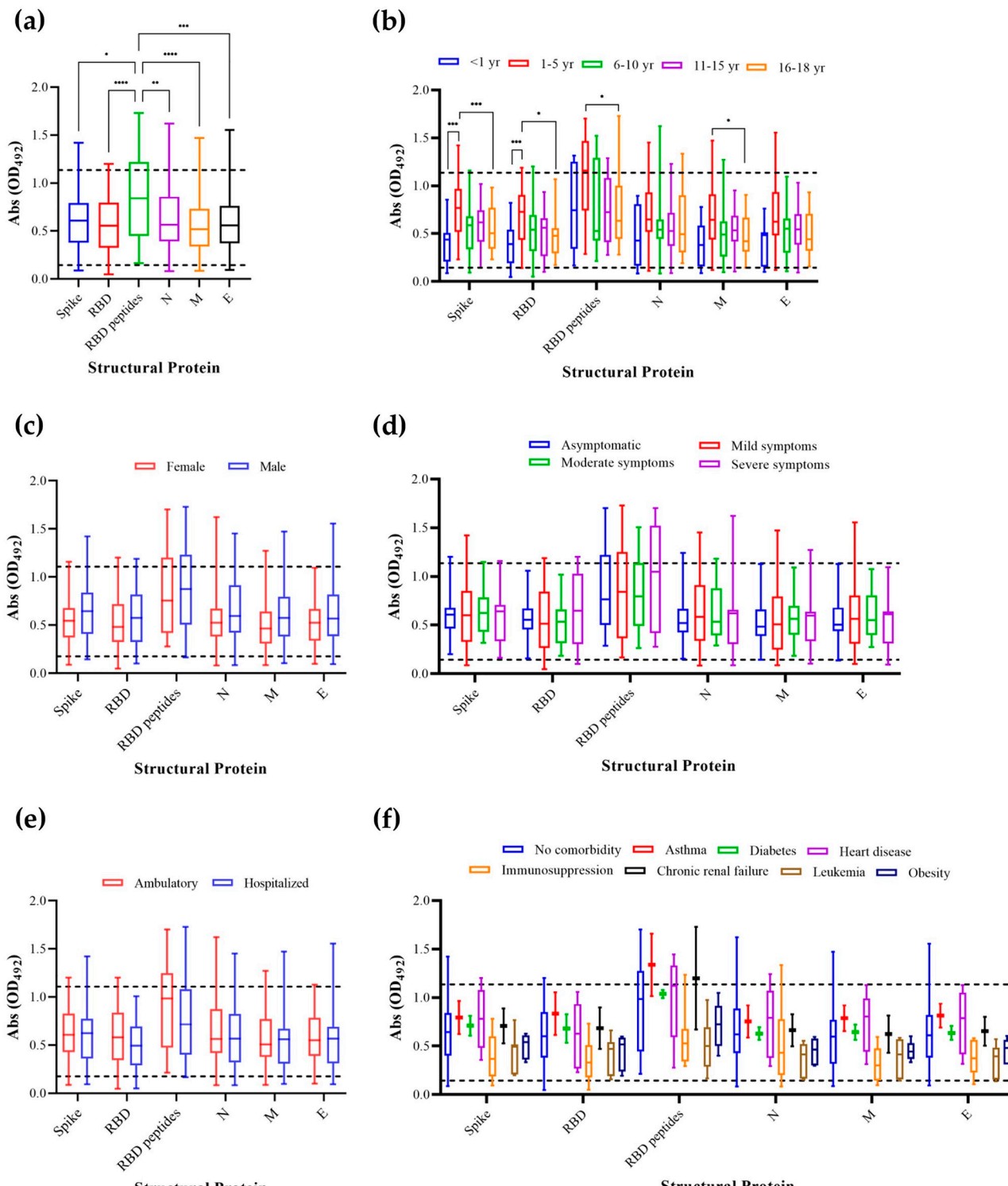

**Fig 2. Evaluation of IgG (whole molecule) antibodies using SARS-CoV-2 S (trimer, RBD, and RBD peptides), N, M, and E proteins in the serum samples of Mexican children and adolescents with COVID-19.** IgG determination of whole molecule IgG antibodies was performed using ELISA in (a) total IgG antibodies, comparisons between IgG and (b) age, (c) sex, (d) severity of illness, (e) patient status, and (f) comorbidity. Bars represent the median of 100 determinations from the serum samples of children and adolescents with COVID-19 performed in duplicate. The top line represents the median value of 10 COVID-19-positive patients, and the bottom line indicates the median value of 10 COVID-19-negative volunteers. Statistical significance was determined using ANOVA followed by the Kruskal–Wallis post hoc test. Statistical significance was considered when $^{*}p \leq 0.05$ to $>0.01$, $^{**}\leq 0.01$ to $>0.002$, $^{***}\leq 0.001$ to $>0.0001$, and $^{***}\leq 0.0001$.

Furthermore, the median Abs values were 0.980 (maximum 1.693; minimum 0.213) for ambulatory and 0.706 (maximum 1.720; minimum 0.160) for hospitalized patients (Fig 2e). The presence of comorbidities did not produce changes in the number of IgG antibodies (Fig 2f). M and E proteins did not produce any significant differences among the most common structural proteins used in antibody-based tests (S and N proteins). Finally, we analyzed the antibody response using γ-chain IgG antibodies and the aforementioned antigens. No differences were observed among all the groups in this population (S1 Fig).

## Discussion

In Mexico, more than three million COVID-19 cases and more than 300,000 deaths have been confirmed, with ~10% corresponding to children and adolescents (https://datos.covid-19.conacyt.mx/). In contrast, in countries such as China, the USA, Italy, and Spain, the prevalence is 1%–2% [8–11]. The factors contributing to the increased incidence of children with COVID-19 in Mexico are unclear, but it has been associated with successful vaccination against SARS-CoV-2 in adults (aged ≥18 years). Considering the scarcity of studies analyzing COVID-19 in children and adolescents, we analyzed the clinical features of Mexican children and adolescents who visited or were admitted to HIMFG in this study. The data presented in this study were obtained from 100 blood samples collected from October 2020 to March 2021. Most patients in this study corresponded to male children with a median age of 9 years, whereas in China, Greece, and the USA, median ages of 6.7 and 11 years have been reported [8,21,24]. COVID-19 occurrence was observed in children aged 1–5 years and adolescents aged 11–15 years, whereas in the USA, the age of COVID-19 occurrence was 15–17 years (32%) and 10–14 years (27%) [8]. This finding is associated with the incidence among those aged 10–14 years (>74,000) and 15–19 years (>152,000) reported by the Secretary of Health in Mexico (https://datos.covid-19.conacyt.mx/). In total, 61% of Mexican children and adolescents were attended as outpatients and 39% were hospitalized, which was consistent with other reports describing a few hospitalized pediatric patients with COVID-19 (26.1%) [4,6,8,21,24]. Hospitalized Mexican children and adolescents at HIMFG were primarily admitted to Pediatric Emergency, whereas other studies have reported the intensive care unit as the major hospital admission in this population [25,26].

All the HIMFG-admitted Mexican children and adolescents recovered and were discharged; however, the national mortality among children aged 0–4 years was more than 460 deaths and that in adolescents aged 15–19 years was 420 deaths, according to the Secretary of Health (https://datos.covid-19.conacyt.mx/). Cardiovascular disease was the major underlying comorbidity observed in this population, which was also reported in another study [22]. Other underlying comorbidities such as leukemia, obesity, and immunosuppression (including HIV-AIDS) were widely distributed among the children and adolescents included in this study, but only two patients were intubated and one of them had preexisting obesity. In Mexico, obesity in children is very common and strongly related to COVID-19 and severe illness [27]. Symptoms such as fever, cough, headache, and odynophagia with sudden onset of symptoms and attack on the general state were primarily reported in this Mexican population; such symptoms are in agreement with those previously reported by Rajakapse et al. [28]. Similar to previous reports, underlying conditions such as asthma and obesity were also associated with severe COVID-19 in Mexican children admitted to HIMFG [8,25,29,30]. In contrast to other studies, in which fever, cough, tachypnea, general malaise, nasal congestion, rhinorrhea, refusal to feed/difficulty feeding, and diarrhea were reported as the most common symptoms, we observed that the children and adolescents without comorbidities had rhinorrhea and polypnea as the major symptoms [31]. Regarding the severity of COVID-19, mild illness was

the most common, followed by moderate illness, which is consistent with previous reports [26]. Nevertheless, other studies have reported asymptomatic COVID-19 as the most common, representing >50% of this population [21,32].

Gaining a better understanding of the antibody response to SARS-CoV-2 is essential for vaccine development, COVID-19 diagnostic and immune responses to SARS-CoV-2, disease susceptibility, and severity determination [17]. ELISA has been used to quantitatively determine SARS-CoV-2-specific antibodies and is widely accepted as an antibody-based diagnostic test [33]. SARS-CoV-2 possesses surface proteins, among which S (binding and fusion), N (viral RNA assembly), E (ionic channel), and M (viral assembly) are the major structural proteins [34–39]. In this study, the major SARS-CoV-2 structural proteins were used to determine the antibody response in Mexican children and adolescents with COVID-19. In CoVs, S and N proteins are more immunogenic than major structural proteins, whereas in SARS-CoV-2, our data showed that all the major SARS-CoV structural proteins exhibited the same ability to detect similar levels of antibodies in Mexican children and adolescents [40]. The RBD peptides used in this study demonstrated the best results in detecting the highest levels of antibodies in this population. As RBD peptides are an *in silico* design of the S1 domain of the SARS-CoV-2 S protein, they have a large potential for being used in antibody-based diagnostic tests (under peer-review). Age was a demographic feature related to the wm IgG antibody levels, and these levels increased with age in Mexican children and adolescents. SARS-CoV-2 S and N protein-specific antibodies have been differentially identified in children and adolescents, and our findings showed that the best antibody response in children aged 1–5 years occurred when using the S trimer, S RBD, and the RBD peptides as antigens [9]. Similar findings were reported in children, adolescents, and young adults using SARS-CoV-2 S and RBD as antigens [41–43]. Furthermore, SARS-CoV-2 M protein, an integral protein comprising 220–260 amino acids, is related to the antibody response in children aged 1–5 years [39]. This protein plays an essential role in viral assembly by interacting with the N protein to encapsulate the SARS-CoV viral RNA [44,45]. Sex, severity of illness, and patient status had a similar antibody response against all SARS-CoV-2 major structural proteins. However, the underlying comorbidities showed low antibody response in children and adolescents with immunosuppression, leukemia, and obesity. These results are in contrast to those of a previous study by Cleto-Yamane et al., in which high seroprevalence, detected using a rapid colloidal gold immunochromatography test, was observed in children and adolescents with immunosuppression and acute lymphoblastic leukemia [46]. Although the effects of obesity in children and adolescents with COVID-19 have not been extensively investigated, the low antibody response observed in the present study may be associated with the chronic subclinical inflammation, impaired immune response, and the underlying cardiorespiratory diseases present in these patients [47].

## Conclusions

The HIMFG, a tertiary care hospital for Mexican children and adolescents, has attended to more than 755 patients with COVID-19 in this specific population. We analyzed the clinical features and antibody response in 100 patients from this population, which mainly comprised 9-year-old boys without comorbidities. They were treated on an outpatient basis (ambulatory) and had mild-to-moderate illness, with the major symptoms being fever, sudden onset of symptoms, cough, headache, attack on the general state, odynophagia, vomiting, shivers, and myalgia. In addition, comorbidities played a crucial role in the antibody response to structural proteins in this population. Children with immunosuppression states, leukemia, and obesity have the lowest antibody response, whereas children aged 1–5 years with heart disease and those without comorbidities had the highest levels of antibody response. This study is one of

the first in the country to focus on the analysis of clinical features in children and adolescents with and without comorbidities and study their association with the antibody response to the major structural proteins of SARS-CoV-2. We also propose further analysis to evaluate the importance of M and E proteins as molecular biomarkers for antibody-based COVID-19 tests in conjunction with S and N proteins.

## Supporting information

**S1 Fig. Evaluation of IgG (gamma chain) antibodies using SARS-CoV-2 S (trimer, RBD, and RBD peptides), N, M, and E proteins in the serum of Mexican children and adolescents with COVID-19.** IgG determination of gamma chain IgG antibodies was performed by ELISA in (a) total IgG antibodies, comparisons between IgG and (b) age, (c) gender, (d) severity of illness, (e) patient status, and (f) comorbidity. Bars represent the median of 100 determinations from the sera of children and adolescents with COVID-19 performed in duplicate. Top line represents the median value of 10 COVID-19-positive patients, and bottom line indicates the median value of 10 COVID-19-negative volunteers. Statistical significance was determined by ANOVA followed by the Kruskal–Wallis post hoc test. Statistical significance was considered when $^*p \leq 0.05$ to $>0.01$, $^{**} \leq 0.01$ to $>0.002$, $^{***} \leq 0.001$ to $>0.0001$, and $^{***} \leq 0.0001$. (TIF)

## Acknowledgments

We thank the research department of Hospital Infantil De Mexico Federico Gomez for all the support given in this study. We thank Q.B.P. Jessica Lizeth Quevedo Moran and Q.B.P. Estefania Ramos Tapia for their technical support given in this study.

## Author Contributions

**Conceptualization:** Karen Cortés-Sarabia, Marcela Salazar-García, Griselda Rodríguez-Martínez, Israel Parra-Ortega, Miguel Klünder-Klünder, Horacio Márquez-González, Adrián Chávez-López, Victor M. Luna-Pineda.

**Data curation:** Israel Parra-Ortega, Miguel Klünder-Klünder, Horacio Márquez-González, Adrián Chávez-López.

**Formal analysis:** Karen Cortés-Sarabia, Armando Cruz-Rangel, Alejandro Flores-Alanis, Samuel Jiménez-García, Israel Parra-Ortega, Miguel Klünder-Klünder, Horacio Márquez-González, Adrián Chávez-López, Victor M. Luna-Pineda.

**Funding acquisition:** Armando Cruz-Rangel, Marcela Salazar-García, Juan Pablo Reyes-Grajeda, Genaro Patiño-López, Oscar Del Moral-Hernández, Berenice Illades-Aguiar, Miguel Klünder-Klünder, Victor M. Luna-Pineda.

**Investigation:** Karen Cortés-Sarabia, Armando Cruz-Rangel, Alejandro Flores-Alanis, Samuel Jiménez-García, Rosa Isela Rodríguez-Téllez, Israel Parra-Ortega, Victor M. Luna-Pineda.

**Methodology:** Karen Cortés-Sarabia, Armando Cruz-Rangel, Alejandro Flores-Alanis, Samuel Jiménez-García, Rosa Isela Rodríguez-Téllez, Berenice Illades-Aguiar, Victor M. Luna-Pineda.

**Project administration:** Marcela Salazar-García, Griselda Rodríguez-Martínez, Juan Pablo Reyes-Grajeda, Victor M. Luna-Pineda.

**Resources:** Karen Cortés-Sarabia, Marcela Salazar-García, Juan Pablo Reyes-Grajeda, Genaro Patiño-López, Israel Parra-Ortega, Oscar Del Moral-Hernández, Berenice Illades-Aguiar, Adrián Chávez-López, Victor M. Luna-Pineda.

**Software:** Genaro Patiño-López.

**Supervision:** Marcela Salazar-García, Victor M. Luna-Pineda.

**Validation:** Karen Cortés-Sarabia, Samuel Jiménez-García, Rosa Isela Rodríguez-Téllez, Victor M. Luna-Pineda.

**Visualization:** Victor M. Luna-Pineda.

**Writing – original draft:** Karen Cortés-Sarabia, Griselda Rodríguez-Martínez, Victor M. Luna-Pineda.

**Writing – review & editing:** Karen Cortés-Sarabia, Victor M. Luna-Pineda.

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
