## [Decision Letter · Decision Letter 0]

13 Jul 2022

PONE-D-22-00992Clinical features and severe acute respiratory syndrome-coronavirus-2 structural protein-based serology of Mexican children and adolescents with coronavirus disease 2019PLOS ONE

Dear Dr. 

Victor Manuel Luna-Pineda

Thank you for submitting your manuscript to PLOS ONE. After careful consideration, we feel that it has merit but does not fully meet PLOS ONE’s publication criteria as it currently stands. Therefore, we invite you to submit a revised version of the manuscript that addresses the points raised during the review process.

I extend an apology for the delay in the review process, unfortunately it was out of our possibilities to do it in a timely manner.

We look forward to receiving your revised manuscript.

Kind regards,

Cecilia Ximenez, Ph.D.

Academic Editor

PLOS ONE

Journal Requirements:

Additional Editor Comments (if provided):

Considering the opinion of the independent reviewer, it would be convenient to discuss why the analysis of neutralizing antibodies was not included

IS A WELL-STRUCTURED ARTICLE WITH A WELL-PLANNED EXPERIMENTAL DESIGN THAT ONLY NEEDS A CAREFUL REVIEW OF THE USE OF LANGUAGE,

Reviewers' comments:

Reviewer's Responses to Questions

**Comments to the Author**

1. Is the manuscript technically sound, and do the data support the conclusions?

Reviewer #1: Yes

2. Has the statistical analysis been performed appropriately and rigorously? 

Reviewer #1: Yes

3. Have the authors made all data underlying the findings in their manuscript fully available?

Reviewer #1: Yes

4. Is the manuscript presented in an intelligible fashion and written in standard English?

Reviewer #1: Yes

5. Review Comments to the Author

Reviewer #1: Nice study addressing the gap of knowledge on antibody responses of children. It would be useful if you could comment on level of correlation of different assays. Also why the neutralization assays were not performed at least to confirm few samples since all the assays are ELISA based.

6. PLOS authors have the option to publish the peer review history of their article (what does this mean?). If published, this will include your full peer review and any attached files.

Reviewer #1: No

---

## [Author Response · Author response to Decision Letter 0]

14 Jul 2022

We really appreciate the editor´s and reviewer´s suggestions made to the manuscript. Below we provide the point-by-point response to comments. We strongly believe that these modifications have improved the manuscript. 

Additional Editor Comments:

Considering the opinion of the independent reviewer, it would be convenient to discuss why the analysis of neutralizing antibodies was not included

IS A WELL-STRUCTURED ARTICLE WITH A WELL-PLANNED EXPERIMENTAL DESIGN THAT ONLY NEEDS A CAREFUL REVIEW OF THE USE OF LANGUAGE,

Reply to editor comments. We have edited the manuscript for language, grammar, structure, and content, from the aspect of fluency and nativity. In addition, we submit the edition certificate from ENAGO Academy. 

Reviewers' comments:

Reviewer's Responses to Questions

Comments to the Author

5. Review Comments to the Author

Reviewer #1: Nice study addressing the gap of knowledge on antibody responses of children. It would be useful if you could comment on level of correlation of different assays. Also why the neutralization assays were not performed at least to confirm few samples since all the assays are ELISA based.

Reply to reviewer 1. 

The aim of this study was to analyze the clinical features and IgG antibodies response against SARS-CoV-2 structural proteins of children and adolescents who visited the Hospital Infantil de México Federico Gómez. The evaluation of neutralizing antibodies in sera sample from SARS-CoV-2 infected patients is part of another study. We want to share part of these studies. Briefly, we have evaluated our MAP8 peptides-based ELISA method for infected SARS-CoV-2, which is under review in Scientific Reports (Cortés-Sarabia et al., SREP-22-00486A). In other study, we have compared the different in-home and commercial (validated) methodologies of S and N proteins based-ELISA (including MAP8 peptides) for determination of IgG antibodies from infected SARS-CoV-2 and vaccinated against COVID-19 sera. In addition, in-home and commercial methods for determination of neutralizing antibodies were correlated with all serology methods using RBD-HRP and soluble hACE2-Fc-based ELISA and viral pseudo particles (VSVG-luc-SARS-CoV-2-S)-based method. These findings showed that the in-house ELISA methods (MAP8 peptides, RBD, and S protein) used in our submitted manuscript have a high correlation with methods for determination of neutralizing antibodies (manuscript in preparation). The SARS-CoV-2 structural proteins (S, N, M and E)-based ELISA method is being also evaluated in a retrospective study using several sera from 2003, 2010 and 2020 to date for validation of our methods (In process).

---

## [Editor Report · Decision Letter 1]

3 Aug 2022

Clinical features and severe acute respiratory syndrome-coronavirus-2 structural protein-based serology of Mexican children and adolescents with coronavirus disease 2019

PONE-D-22-00992R1

Dear Dr. Victor Manuel Luna-Pineda

We’re pleased to inform you that your manuscript has been judged scientifically suitable for publication and will be formally accepted for publication once it meets all outstanding technical requirements.

Kind regards,

Cecilia Ximenez, Ph.D.

Academic Editor

PLOS ONE
---

## [Editor Report · Acceptance letter]

5 Aug 2022

PONE-D-22-00992R1 

Clinical features and severe acute respiratory syndrome-coronavirus-2 structural protein-based serology of Mexican children and adolescents with coronavirus disease 2019 

Dear Dr. Luna-Pineda:

I'm pleased to inform you that your manuscript has been deemed suitable for publication in PLOS ONE. Congratulations! Your manuscript is now with our production department. 

Kind regards, 

on behalf of

Dr. Cecilia Ximenez 

Academic Editor

PLOS ONE